# Atmospheric radiative profiles during EUREC$^4$A

Anna Lea Albright[1,*], Benjamin Fildier[2,*], Ludovic Touzé-Peiffer[1,*], Robert Pincus[3,4], Jessica Vial[1], and Caroline Muller[2]

[1]Laboratoire de Météorologie Dynamique, Sorbonne University, Paris, France.
[2]Laboratoire de Météorologie Dynamique, Ecole Normale Supérieure, Paris, France
[3]Cooperative Institute for Research in Environmental Sciences, University of Colorado, Boulder, CO, USA
[4]NOAA Physical Sciences Lab, Boulder, CO, USA
[*]These authors contributed equally to this work.

**Correspondence:** Anna Lea Albright (anna-lea.albright@lmd.jussieu.fr)

**Abstract.**

The couplings among clouds, convection, and circulation in trade-wind regimes remain a fundamental puzzle that limits our ability to constrain future climate change. Radiative heating plays an important role in these couplings. Here we calculate clear-sky radiative profiles from 2504 *in situ* soundings (1068 dropsondes and 1436 radiosondes) collected during the EUREC$^4$A field campaign, which took place in the downstream trades of the western tropical Atlantic in January-February 2020. We describe the method used to calculate these cloud-free, aerosol-free radiative profiles. We then present preliminary results sampling variability at multiple scales, from the variability across all soundings to groupings by diurnal cycle and mesoscale organization, as well as individual soundings associated with elevated moisture layers. We also perform an uncertainty assessment and find that the errors resulting from uncertainties in observed sounding profiles, and ERA5 reanalysis employed as upper and lower boundary conditions are small. The present radiative profile data set can provide important additional detail missing from calculations based on passive remote sensing and aid in understanding the interplay of radiative heating with dynamic and thermodynamic variability in the trades. The data set can also be used to investigate the role of low-level radiative cooling gradients in generating shallow circulations. All data are archived and freely available for public access on AERIS (Albright et al. (2020), https://doi.org/10.25326/78).

## 1  Introduction

The EUREC$^4$A field campaign, which took place in January and February 2020 in the downstream trades of the western tropical Atlantic, was designed to elucidate the couplings among clouds, convection, and circulation in trade-wind regimes and understand the role of this interplay in climate change (Bony et al., 2017). Shallow trade-wind clouds cover large parts of tropical oceans (Wood, 2012), yet their response to warming remains largely unknown, and uncertainty in shallow convective processes are the cause for large uncertainties in climate projections (Bony and Dufresne, 2005; Vial et al., 2013; Sherwood et al., 2014; Zelinka et al., 2020). Among all physical processes involved in shallow convection, atmospheric radiative cooling emerges as key to the coupling between low-level circulations and convection. Understanding the dynamics driven by varia-

tions in radiative heating rates, and potential relationship to the mesoscale organization of clear and cloudy regions, was one motivation for the campaign (Bony et al., 2017).

A characteristic feature of the trade-wind vertical moisture profile is a sharp humidity gradient between the moist marine boundary layer and dry, subsiding free troposphere around two kilometers Riehl et al. (1951); Malkus (1958). This characteristic vertical moisture structure has important implications for radiative cooling profiles, but it is difficult to observe with satellite remote sensing (Stevens et al., 2017). Indeed, moisture profile features, such as the sharp decreases in moisture at the top of the marine boundary layer or elevated moisture layers, are smaller than typical weighting functions of even hyper-
spectral instruments (e.g. Maddy and Barnet, 2008; Schmit et al., 2009; Menzel et al., 2018), especially in the lowest three kilometers, corresponding to the weakest absorption lines (Chazette et al., 2014). The lack of informative observations means that the vertical profile of water vapor in large-scale atmospheric analyses do not represent the fine-scale moisture structure indicated by soundings (Pincus et al., 2017). Errors in the vertical moisture structure estimated from passive remote sensing produce corresponding errors in radiative cooling profiles computed from retrievals and/or analyses, making *in situ* soundings
especially valuable.

      Here we calculate radiative profiles from 2504 soundings (1068 from dropsondes and 1436 from radiosondes) collected during EUREC[4]A, whose network of observations provided extensive sampling of the tropical trade-wind environment. Similar studies have been conducted over continents as part of the Atmospheric Radiation Measurement program (Kato et al., 1997; Mlawer et al., 1998), over the western Pacific warm-pool region as part of the Coupled Ocean Atmosphere Response
Experiment (Guichard et al., 2000), and over the western tropical Atlantic, albeit focused on transported Saharan dust layers (Gutleben et al., 2019). The present radiative profiles have the potential to complement and further what can be learned from calculations based on passive remote sensing. In addition, this data set may help in understanding how low-level gradients in radiative cooling fuel shallow circulations, as observed to emerge in remote sensing and large eddy simulations (L'Ecuyer et al., 2008; Stephens et al., 2012; Seifert et al., 2015). These shallow circulations are speculated to influence the mesoscale
spatial organization of shallow convection, a question at the core of EUREC[4]A (Bony et al., 2020; Stevens et al., 2020).

      In Section 2, we describe the data, the radiative transfer code, and the procedure underlying the calculation of the radiative profiles. We then present initial results to open the discussion on questions that could be explored with these radiative profiles (Section 3). Lastly, we perform an uncertainty assessment (Section 4) and find that errors resulting from uncertainties in the sea surface skin temperature, *in situ* soundings, and ERA5 reanalysis used as boundary conditions are modest.

## 50  2    Data and methods

### 2.1    Radiosonde and dropsonde data

From January 8 to February 19, over 2500 atmospheric soundings were conducted using dropsondes and radiosondes over the western tropical Atlantic ocean south and east of Barbados. As the sondes fall or ascend, their simple autonomous sensors, equipped with a GPS receiver, measure the vertical profiles of pressure, temperature, relative humidity, and instantaneous hor-
izontal wind. To calculate radiative profiles, we employ level-3 data, which have been interpolated into a common altitude grid

with 10 meter spacing (Stephan et al., 2020; George et al., 2020). We select dropsondes and radiosondes that have measurements on more than ten atmospheric levels in total. This filter suffices to remove failed soundings and results in an input data set consisting of 1068 atmospheric profiles from dropsondes and 1436 profiles from radiosondes. The minimum and maximum levels $z_{min}$ and $z_{max}$ measured by each sonde are reported in the final data set.

Figure 1a shows the geographic and temporal distributions of the sondes used to calculate the radiative profiles. Radiosondes were launched from a network of one land station and four research vessels, within a region ranging from 51–60°W to 6–16°N. On land, radiosondes were launched from the Barbados Cloud Observatory (BCO), located on a promontory on the easternmost point of Barbados called Deebles Point (13.16°N, 59.43°W), where it is exposed to relatively undisturbed easterly trade-winds. The fleet of four research vessels includes the French research vessel L'Atalante, two German research vessels Maria S. Merian (MS-Merian) and Meteor, and the American research vessel from the National Oceanic and Atmospheric Administration (NOAA) Ronald H. Brown (RH-Brown). Dropsondes were launched from both the German High Altitude and Long Range Research Aircraft (HALO) and the United States Lockheed WP-3D Orion from NOAA (WP-3D). HALO typically flew at an altitude of 30,000 ft (approximately 9 km), following a circular flight pattern with 90 km radius centered at 13.3°N, 57.7°W. When launching sondes, the WP-3D flew at 24,000 ft (approximately 7 km), releasing sondes along both linear and circular patterns in the region covered by HALO, as well as further to the east close to the nominal position of the RH-Brown.

Radiosondes were launched every four hours, daily from January 8–February 19, 2020, approximately synchronously from each platform. Given that the time-lag between ascending and descending radiosondes is on the order of hours, and that there is substantial horizontal drift between the ascent and descent, we chose to compute separate radiative profiles for ascending and descending radiosondes. For dropsondes, HALO flight takeoffs were staggered at 5, 8, and 11 am local time, with flights lasting approximately eight hours, yielding roughly 72 sondes per flight. The WP-3D undertook three night flights, which allows for a better characterization of the diurnal cycle, together with the radiosondes launched during the night (Figure 1b).

We refer the reader to Stephan et al. (2020) and George et al. (2020) for a complete description of the radiosonde and dropsonde data sets, respectively, and Bony et al. (2017) and Stevens et al. (2020) for an overview of the campaign scientific motivations and measurement strategy.

## 2.2  Radiative transfer calculation

The radiative transfer code used here, RRTMGP (Rapid Radiative Transfer Model for GCMs, Parallel) (Pincus et al., 2019), is a plane-parallel correlated-$k$ two-stream model based on state-of-the-art spectroscopic data for gas and condensate optics. It is based on line parameters from Atmospheric and Environmental Research and the MT_CKD water vapor continuum absorption model (Mlawer et al., 2012). The calculation of radiative profiles from radiosonde and dropsonde data then proceeds in the following way:

1. vertical soundings of temperature, pressure, and water vapor specific humidity at 10 meter resolution are interpolated onto a 1 hPa vertical grid and then merged with temperature and specific humidity from ERA5 reanalyses in the following manner. Sonde measurements below 40 m are first truncated for all sondes: radiosondes do not provide data in this surface

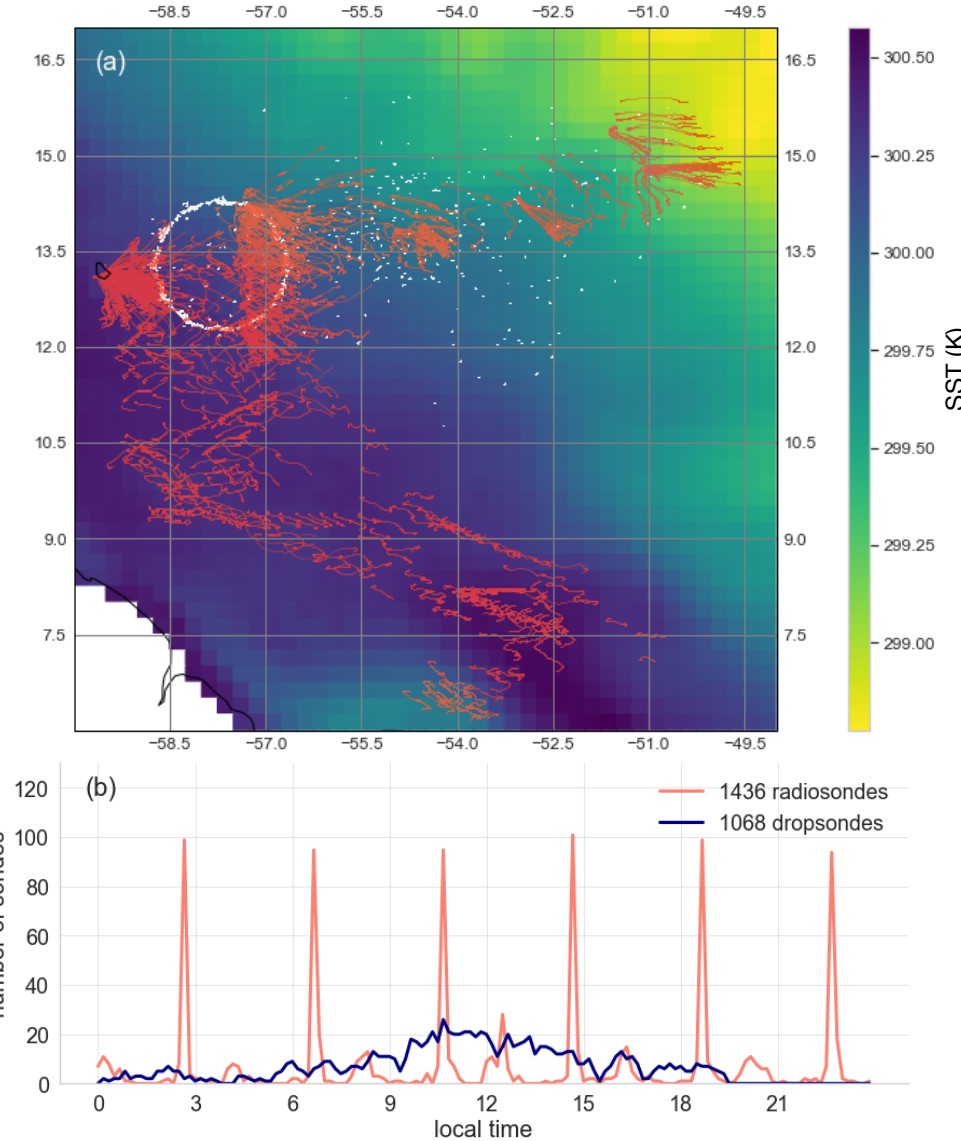

**Figure 1.** (a) The EUREC[4]A sounding network: 1068 soundings from dropsondes (white) and 1436 from radiosondes (coral). We employ 807 dropsondes launched from HALO and 261 dropsondes from the WP-3D to calculate radiative profiles, as well as 276, 342, 147, 362, and 309 radiosondes launched from the Atalante, BCO, MS-Merian, Meteor, and RH-Brown, respectively. Background colors show sea surface skin temperature ($SST_{skin}$) from ERA5 reanalysis at $0.25°$ resolution averaged over January and February. (b) The diurnal distribution of the 1068 dropsondes (blue) and 1436 radiosondes (coral) with sonde launch-time binned in 10-minute intervals.

layer because of deck heating effects on ships (Stephan et al., 2020), and we apply the same filter to dropsondes for consistency. The ERA5 profiles at hourly and 0.25° resolution (European Centre for Medium-Range Weather Forecasts, 2017) are linearly interpolated temporally and spatially to the time, latitude, and longitude of the sounding. ERA5 values are used above the highest level measured by each sonde to extend the observed soundings vertically to 0.1hPa and account for the effect of high-altitude thermodynamic variability on the radiative cooling profiles below. To obtain the lower boundary condition, we linearly interpolate the ERA5 sea surface skin temperature ($SST_{skin}$), also at hourly and 0.25° resolution (European Centre for Medium-Range Weather Forecasts, 2017), to the time, longitude and latitude where the sounding was launched;

2. $CO_2$ concentrations are set to the present day value of 414 ppm while $CH_4$, $O_3$ and $N_2O$ concentrations are taken from the standard tropical atmosphere profile of Garand et al. (2001);

3. the set of resulting profiles is then used as input to RRTMGP to derive upwelling and downwelling clear-sky radiative fluxes in the shortwave and longwave ranges of the spectrum. The calculation uses a spectrally-uniform surface albedo of 0.07 and a spectrally-uniform surface emissivity of 0.98, typical values for tropical oceans.

Dropsondes and radiosondes drift horizontally as they rise and/or fall (Figure 1a), which could lead to slight errors due to aliasing of horizontal variability in moisture content into vertical variability. This potential source error is less pronounced for dropsondes than for radiosondes due to their faster speed of travel through the troposphere.

We compute radiative fluxes and heating rates only for the gaseous component of the atmosphere, without explicitly taking into account cloud or aerosol properties. These radiative profiles are therefore clear-sky and aerosol-free. The soundings do, however, capture the water vapor structure, including regions of high humidity in cloud areas and aerosol layers. Cloud cover in trade-wind regimes is relatively low, between 10% (Nuijens et al., 2015) and 20% (Medeiros and Nuijens, 2016) for active clouds, so cloud-free, or clear-sky, profiles are representative of the thermodynamic environment. Taking into account the influence of cloud liquid water would require a number of *ad hoc* assumptions about microphysical and optical properties within clouds (see for instance Guichard et al., 2000). Similarly, we do not directly represent the radiative effect of mineral dust aerosols. The dominant aerosol radiative effect in this region has been shown to result from the covariance of aerosols with water vapor, such that aerosols tend to be associated with elevated moisture layers Gutleben et al. (2019, 2020). Dust aerosol layers are, moreover, more common in the summer than in winter (Lonitz et al., 2015). We leave open the possibility that direct scattering by dust aerosols has an additional role on radiative heating rates, but do not have the coincident data to appropriately address this question for all soundings.

## 3 Preliminary results and discussion

This section includes a first exploration of the data set. We examine radiative variability at different scales – across all soundings, at the diurnal timescale, and according to different patterns of mesoscale organization – as well as in individual profiles showing the influence of sharp vertical moisture gradients on radiative heating rates.

## 3.1 Variability across soundings

A distribution of longwave, shortwave, and net heating rates, as well as large-scale thermodynamic quantities, are shown in Fig. 2. Local extrema in the median shortwave, longwave, and net heating rates occur near 2 km (Fig. 2d,e,f), associated with the rapid decrease in specific and relative humidity at this level (Fig. 2b,c). The top of the planetary boundary layer, or interface between the moist marine boundary layer and dry free troposphere above, is expected to occur around 2km in the trades (Malkus, 1958; Cao et al., 2007; Stevens et al., 2017). The spread in specific and relative humidity is greater than that in temperature, suggesting a strong role for moisture variability on the variability in radiative heating rates. On average, longwave cooling is stronger than shortwave heating, such that net heating rates are largely negative from the surface up to 10 km, with a median value around -1 K/day. Additional local minima in longwave heating are observed around 3 and 5 km between the 5% and 25% quantiles. These local minima could, for instance, correspond to the radiative effect of elevated moisture layers arising from convection detraining moisture at these higher levels, albeit less frequently, or aerosol layers associated with increased water vapor concentrations (Stevens et al., 2017; Wood et al., 2018a, b; O et al., 2018; Gutleben et al., 2019).

We next partition radiative heating variability into its variability in time (e.g. diurnal cycle, day-to-day variability) and regarding the spatial characteristics of the convection field (e.g. the spatial distribution of clear and cloudy regions).

## 3.2 Diurnal cycle and day-to-day variability

Figure 3 gives an overview of the diurnal variability of radiative heating, which has been implicated in the diurnal cycle of convection and cloudiness (e.g., Gray and Jacobson Jr, 1977; Randall and Tjemkes, 1991; Ruppert and Johnson, 2016). Short-wave radiative heating follows the solar cycle. Longwave heating rates show less diurnal variability and have approximately the same amplitude (with an opposite sign) as shortwave heating rates during daytime. This compensation between longwave cooling and shortwave heating results in a daytime net heating rate that is slightly positive in the lower 2km. The daytime heating contributes to stabilizing the lower atmosphere, disfavoring convection. At night, strong radiative cooling destabilizes the lower troposphere and strengthens convection. The maximum nighttime longwave cooling occurs slightly above 2 km, with secondary cooling peaks occurring around 4 and 6km. During daytime, the peak in stabilizing radiative heating appears slightly below 2km. This difference in the height of peak radiative heating, albeit of different sign, could reflect differences in the height of the moist, convecting layer over the diurnal cycle: a shallower marine boundary layer during the day that deepens at night (Vial et al., 2019). These considerations highlight the potential for subtle interactions among radiation, convection, and cloudiness on the diurnal timescale.

Fig. 4 shows the day-to-day evolution of the shortwave (top), longwave (middle) and net (bottom) heating rates derived from radiosondes launched at BCO. In the shortwave and net heating rates, the daily stripes are due to zero shortwave heating during the night. In the longwave component alone, the amplitude of the diurnal cycle is less evident. Regarding the day-to-day variability, both in the shortwave and the longwave components, trends in the height-evolution of the radiative heating maxima appear to persist over several days. These trends are likely due to variations in humidity (e.g. Dopplick, 1972; Jeevanjee and Fueglistaler, 2020) and are consistent with the presence of multi-day trends in moisture observed at BCO during the campaign

(see Figure 13 in Stevens et al., 2020). At the end of the campaign, the rise in the peak of longwave cooling appears to

correspond to the rising location of the interface between the moist, convecting layer below and dry free troposphere above (not shown). The persistence and evolution of radiative heating patterns could be tied to larger-scale synoptic moisture activity or to the evolution of mesoscale organization patterns.

### 3.3    Radiative signatures of mesoscale patterns of cloud organization

We next aggregate radiative heating rates spatially. Fig. 5 illustrates four representative cases of the Fish-Gravel-Flower-Sugar

classification established previously for mesoscale (20-2,000km) organization patterns of clear and cloudy regions (Bony et al., 2020; Stevens et al., 2020). These cloud organization patterns were identified visually from satellite imagery and correspond to differences in large-scale environmental conditions (Bony et al., 2020). They are also observed to have different top-of-the-atmosphere radiative effects (Bony et al., 2020). As outlined in Stevens et al. (2020), Sugar refers to a 'dusting' of small, shallow clouds with low reflectivity and a random spatial distribution. Gravel clouds tend to be deeper than Sugar (up to

3-4km), have little stratiform cloudiness, precipitate, and organize along apparent gust fronts or cold pools at the 20-200km scales. Fish are skeletal networks (often fishbone-like) of clouds at the 200-2,000km scale with stratiform cloud layers; the Fish pattern is often associated with extratropical intrusions. Flowers are circular features defined by their stratiform cloud elements. Both Fish and Flowers are surrounded by large swaths of clear air.

We choose four days as an example of the large-scale environmental and radiative signature of each pattern, given the

spatial pattern observable in the GOES-16 satellite images in the HALO flight path shown by the white circle. We plot daily-mean profiles for temperature, specific humidity, and relative humidity (Fig. 5a,b,c), as well as shortwave, longwave, and net radiative heating rates (Fig. 5d,e,f). These profiles were calculated from approximately 70 HALO dropsondes launched during the eight-hour flight on each day. We also plot the standard deviation of radiative heating for each flight (Fig. 5g,h,i). As a first approximation, the standard deviation of daily radiative heating profiles acts as a proxy for spatial variability in radiative

heating rates.

Spatial variability in radiative heating has been shown to drive shallow circulations (e.g. Naumann et al., 2019) and affect convective organization (e.g. Bretherton et al., 2005; Muller and Held, 2012). In this illustrative example, the differences in the mean and standard deviation of the radiative heating rates hint at a role for differences in radiative cooling rates in the onset or maintenance of mesoscale patterns of organization. For instance, the 'Fish' pattern on January 22, 2020 is associated

with a moister lower troposphere between 1 and 3km and slightly drier free troposphere above 4km. This vertical moisture distribution may give rise to the observed vertical variability in radiative heating rates, with larger peaks in the mean profile (Fig. 5e) and standard deviation (Fig. 5h) in radiative heating observable between 2 and 4km, likely corresponding to strong humidity gradients at these levels.

### 3.4    Effect of sharp moisture gradients on radiative heating profiles

Figure 6 highlights the radiative signatures of elevated moisture layers, which can persist for multiple hours at inversion levels (Stevens et al., 2017; Wood et al., 2018a; Gutleben et al., 2019). We focus in detail on two thermodynamic and radiative heating

profiles of a particular elevated moisture layer extending to 4 kilometers, alongside GOES-16 images (Fig. 6i,j) corresponding to these soundings. This structure persisted for at least four hours on January 24, 2020, and we plot thermodynamic conditions and radiative heating profiles sampled three hours apart, at 12:55 and 15:55 UTC (see Fig. 6). A striking feature is the sharp peak in longwave cooling at the top of the moisture layer of nearly -20 K/day at 15:55 UTC, corresponding to the strong humidity gradient, with relative humidity decreasing by nearly 70% in 100 meters (Fig. 6c,d).

Although we calculate clear-sky profiles only, the present work could be extended to account for radiative effect of cloud liquid water, which could be used, for instance, to investigate the radiative effect of geometrically- and optically-thin 'veil clouds' persisting at inversion levels (Wood et al., 2018a, b; O et al., 2018), such as those illustrated by the flight photographs (Fig. 6a,e). Over global oceans, approximately half of low clouds do not fully attenuate space-borne lidar, suggesting that these optically-thin clouds contribute significantly to total cloud cover estimates (Leahy et al., 2012) and could have an important radiative impact (e.g., Wood et al., 2018b).

## 4   Uncertainty assessment

To evaluate the robustness of our results and ensure good use of this data set, we performed several uncertainty assessments by perturbing the $SST_{skin}$, *in situ* moisture data, and ERA profiles used. We also included in the data set the minimum and maximum levels $z_{min}$ and $z_{max}$ measured by each sonde. Unless indicated otherwise, the errors reported below correspond to a subset of profiles with valid data starting at 40 m (ie. $z_{min} \leq 40$ m) and during daytime, which corresponds to 1314 profiles. The daytime filter was required for relevant calculation of the error in the shortwave, and then kept for consistency for the longwave, but the magnitude of errors in the longwave is not affected by this filter (not shown).

We first test the sensitivity to the ERA5 $SST_{skin}$. To this end, we perturbed the original $SST_{skin}$ by $\pm 0.42$ K and recalculate all heating rates. This value is chosen as it corresponds to the root-mean-square-error (RMSE) between between ERA5 $SST_{skin}$ and Marine-Atmosphere Emitted Radiance Interferometers (M-AERI) measures taken during a series of cruises in the Carribbean Sea from 2014 to 2019 (Luo and Minnett, 2020). Figure 7 shows the RMSE between the original and perturbed radiative profiles (blue curves). In the longwave and net, the effect of the perturbation is strong in the first atmospheric layer, but then decreases rapidly and becomes negligible after a few hundred meters. Except for the first few atmospheric layers, the uncertainty around the $SST_{skin}$ can therefore be safely neglected.

We then investigate the sensitivity to the uncertainty of sounding measurements by perturbing all soundings by a vertically-uniform relative error and redoing all radiative transfer calculations. The manufacturer predicts an uncertainty of $\pm 0.1$ K for the temperature and $\pm 3$ % for specific humidity (Vaisala, 2018). The temperature uncertainty has virtually no effect on radiative profiles (not shown). The effect of $\pm 3$ % uncertainty on the specific humidity profiles is shown in Fig. 7 in red. The highest RMSE for this specific humidity perturbation occurs in the cloud layer, between 800 m and 2 km, with a magnitude of 0.05 K/day for net radiative heating. A secondary peak with a magnitude of 0.03 K/day is also evident near the inversion, at about 3 km. Given a median radiative heating value of -1K/day throughout the lower troposphere (Sec. 3.1), these errors are roughly 3-5% for the net radiative heating. These maxima likely correspond to the cumulative errors at the altitude of large

vertical humidity gradients, which lead to peaks in longwave, and to a lesser extent shortwave heating rates for individual profiles.

Finally, we explore the uncertainty associated with ERA5 temperature and humidity data employed as an upper boundary condition. Similarly to the uncertainty analysis for the sounding data, we perturb ERA5 3D fields – used as input to the radiative transfer code – by a uniform relative error. Previous studies have shown that ERA5 reanalyses can present biases of various kinds (Nagarajan and Aiyyer, 2004; Dyroff et al., 2015). We compare ERA5 humidity and temperature data with coincident radiosonde measures to obtain an estimate of ERA5 biases up to 100 hPa. From the surface to 100 hPa, the RMSE in temperature between co-located radiosonde soundings and ERA5 is between 0.3 and 0.7 K, with a mean of 0.5 K, and between 5% (at the surface) and 70 % (near the inversion) for the specific humidity, with a mean around 30 %.

Fig. 7 only shows the effect of the ERA5 specific humidity uncertainty, taken at $\pm$ 30 %, on radiative profiles, as the temperature has once again a negligible influence. The corresponding green curves (respectively dashed and solid) reveal local maxima in the longwave and net heating rates around 3, 7 and 9.5 km. Again given a median radiative heating value of -1K/day throughout the lower troposphere (Sec. 3.1), the errors at these local peaks are between 10-30%. These maxima coincide with the modes in the frequency distribution of the highest level $z_{max}$ measured by the soundings, indicated in grey in the left panel. These peaks suggest that the uncertainty arises from the large discontinuities emerging at the ERA5-sounding junction level when perturbing ERA5 humidity profiles. The results suggest that the corresponding uncertainty mainly occurs in the vicinity of the junction levels. This notion is further confirmed by calculating the RMSE only on profiles which have data between 40 m and 8 km (ie. $z_{min} \leq 40$ m and $z_{max} \geq 8$ km, dotted green curve): the remaining 1117 profiles left do not contain vertical discontinuities in humidity in this range, and we see that the remaining upper-tropospheric discontinuities do not affect heating rates in the lowest troposphere.

Overall, the small uncertainty values given with these tests support the robustness of this data set and gives confidence regarding its use for more detailed investigations in the lower troposphere. The uncertainty from sea surface skin temperature is limited to the first few atmosphere layers, and uncertainty from merging with ERA5 specific humidity is largely contained to the sounding-reanalysis junction point. Uncertainty associated with observed specific humidity profiles produces localized errors in the cloud and inversion layers below 3km, though these errors are approximately 5% or less. We recommend that users carefully compare the magnitude of the signal they analyze with the magnitudes of the errors provided here.

## 5   Conclusions

The first objective of this work is to present the method used to calculate clear-sky, aerosol-free radiative profiles from 2504 radiosonde and dropsonde soundings launched during the EUREC[4]A field campaign. These radiative profiles are calculated using a state-of-the-art correlated-$k$ model, RRTMGP, in which ERA5 reanalyses provide lower and upper boundary conditions. We then aggregate the radiative heating profiles at multiple scales to examine temporal and spatial variability in trade wind regimes. We find that radiative heating rates in the wintertime trade-wind environment display significant diurnal and day-to-day variability, and we observe hints that this variability may be associated with different types of mesoscale organization. An

uncertainty assessment is further conducted to demonstrate that the influence of uncertainties in the sounding data, and upper and lower boundary conditions, is small relative to the magnitude of estimated radiative heating.

These results present a first overview of how this data set could help answer existing research questions, in particular: 1) What is the role played by radiation in the mesoscale organization of shallow convection? (e.g. Seifert and Heus, 2013; Bretherton and Blossey, 2017) 2) what is the interplay between the diurnal variability in radiative heating, convection, and cloudiness? (e.g., Gray and Jacobson Jr, 1977; Ruppert Jr and O'Neill, 2019; Vial et al., 2019), and 3) what is the influence of clear-sky radiative cooling gradients on atmospheric circulations? (e.g. Gray and Jacobson, 1977; Mapes, 2001; Emanuel et al., 2014; Thompson et al., 2017; Naumann et al., 2019). Such questions regarding the coupling of clouds, convection, and circulations in trade-wind regimes are at the heart of the EUREC$^4$A field campaign, and the radiative profiles presented here complement other EUREC$^4$A observations and data products in forming a toolbox for these investigations.

## 6   Code and data availability

All data are archived and freely available for public access on AERIS (Albright et al., 2020, https://doi.org/10.25326/78). The code used to compute the radiative profiles and python scripts used to generate the figures of the present paper are publicly released on Zenodo (https://doi.org/10.5281/zenodo.4010195) and Github (https://github.com/bfildier/Albright2020).

*Author contributions.*   A.L.A, B.F., and L.T.P contributed equally to the analysis, figures, and text. R.P., C.M. and J.V. helped in conceptualizing and guiding this project and contributed to the manuscript.

*Competing interests.*   The authors declare that no competing interests are present.

*Acknowledgements.*   A.L.A is grateful for support from the European Research Council (ERC) under the European Union's Horizon 2020 research and innovation programme (grant agreement #694768). B.F. and C.J.M. gratefully acknowledge funding from the European Research Council (ERC) under the European Union's Horizon 2020 research and innovation programme (Project CLUSTER, grant agreement #805041). L.T.P gratefully acknowledges the funding of his PhD by the AMX program of the Ecole Polytechnique. R. P. is grateful for support from the NOAA Climate Program Office program on Climate Variability and Predictability. To access ERA5 reanalysis data, this study benefited from the IPSL Prodiguer-Ciclad facility which is supported by CNRS, UPMC, Labex L-IPSL and funded by the ANR (Grant #ANR-10-LABX-0018) and by the European FP7 IS-ENES2 project (Grant #312979).

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

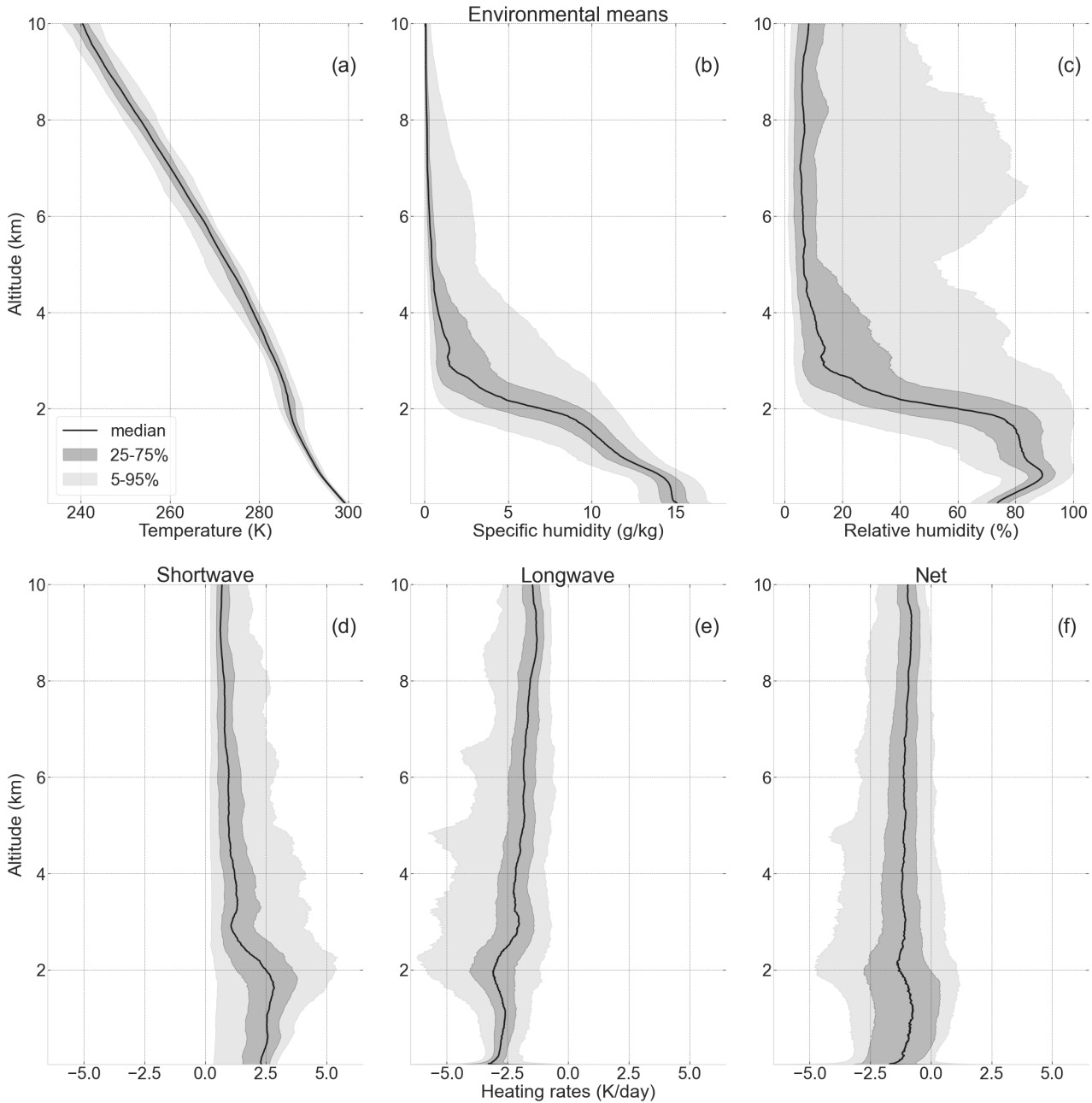

**Figure 2.** Top: Temperature (a), specific humidity (b) and relative humidity (c) (with respect to ice for $T < 0°C$) from EUREC$^4$A dropsonde and radiosonde data. Bottom: Shortwave (d), longwave (e) and net (f) heating rates calculated from EUREC$^4$A dropsonde and radiosonde data using the radiative transfer code RRTMGP. The center traces are the median profiles, and the medium and light grey shadings indicate the 25–75% and 5–95% intervals, respectively. For the shortwave, the median and the interquartile range are calculated using daytime values only.

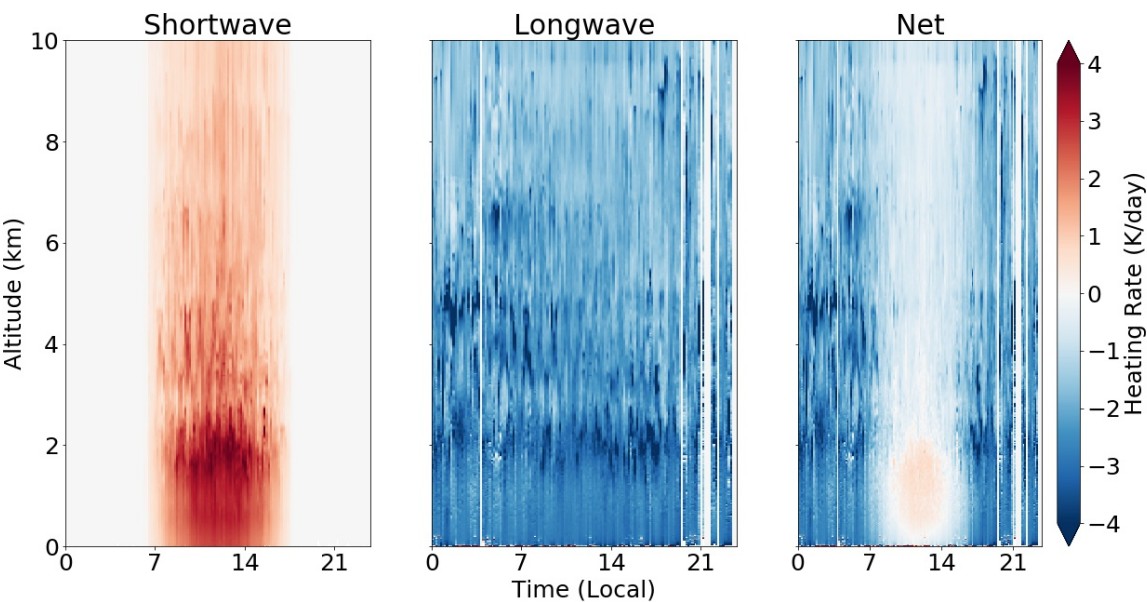

**Figure 3.** Diurnal composite of shortwave (left), longwave (middle), and net (right) clear-sky heating rates binned in 10-minute intervals. Colored shadings indicate heating rates in units of K/day. The data are plotted with respect to local solar time to simplify interpretation of the diurnal cycle. White indicates the absence of data. We note that some variability, such as in the nighttime longwave radiative cooling variability, could result from different numbers of sondes launched throughout the diurnal cycle (as illustrated in Fig. 1b).

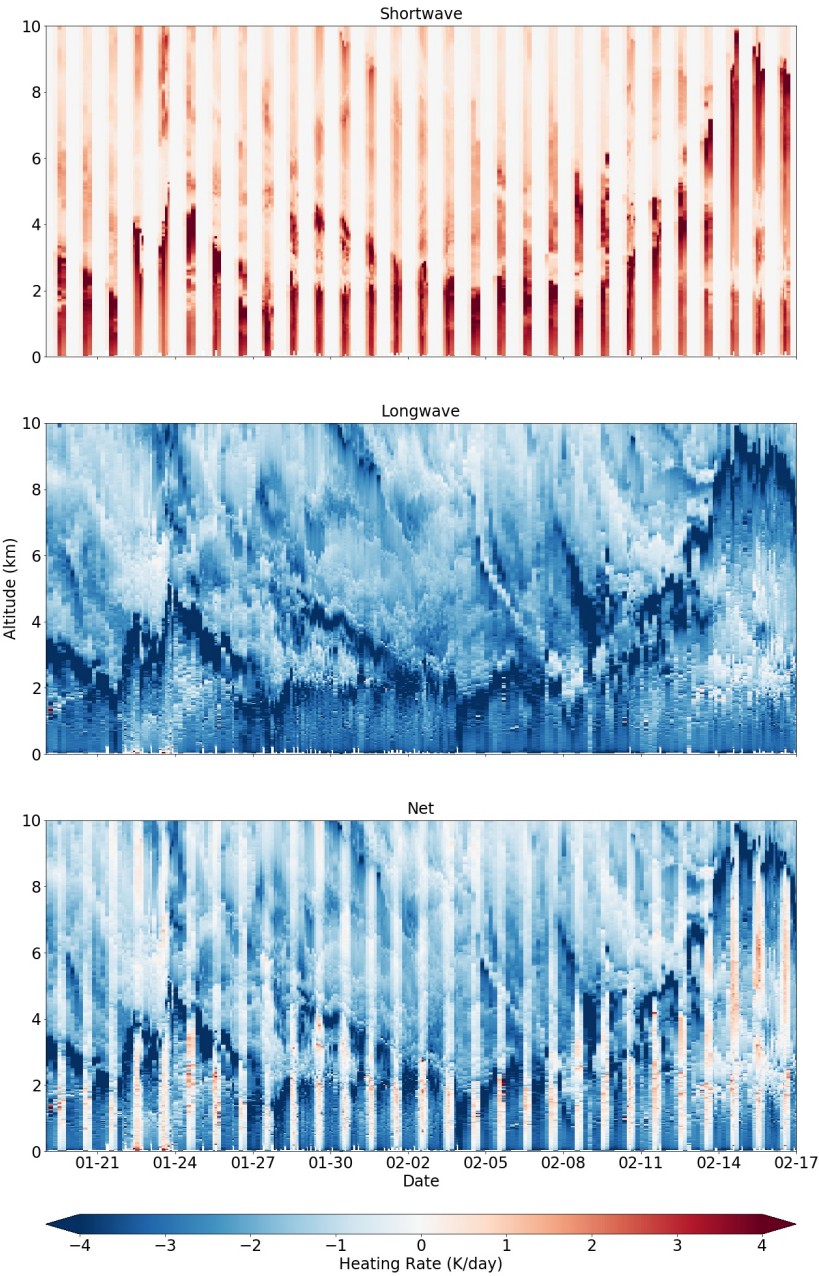

**Figure 4.** Shortwave (top), longwave (middle), and net (bottom) heating rates at BCO during EUREC[4]A, from January 19 to February 17. The heating rates are calculated from radiosondes launched at BCO. In colors are heating rates with units of K/day. White indicates the absence of data.

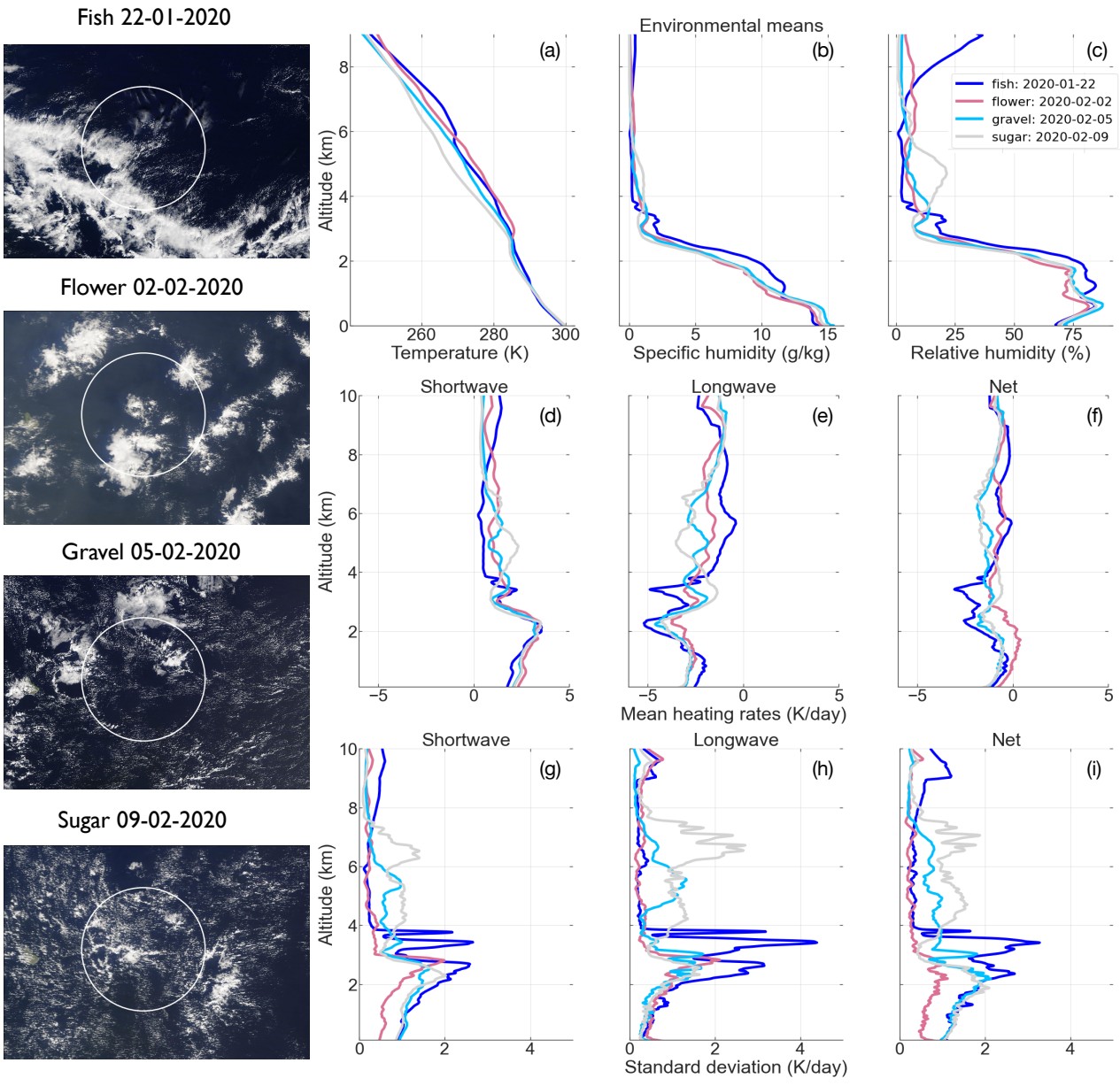

**Figure 5.** Thermodynamic (a-c), daily mean radiative heating (d-f), and daily standard deviation of radiative heating (g-i) profiles classified by mesoscale organization pattern, using a characteristic example of each type as diagnosed from snapshots from GOES-16 infrared channel (left column). This figure employs HALO dropsondes launched in the circular flight pattern (shown by the white circle) on the chosen day, corresponding to roughly 70 dropsondes each. We focus on the spatial extent of the HALO flight pattern because the cloud organization pattern does not necessarily extend across the entire sampling domain Figure 1a, nor have the patterns been shown to be scale-invariant.

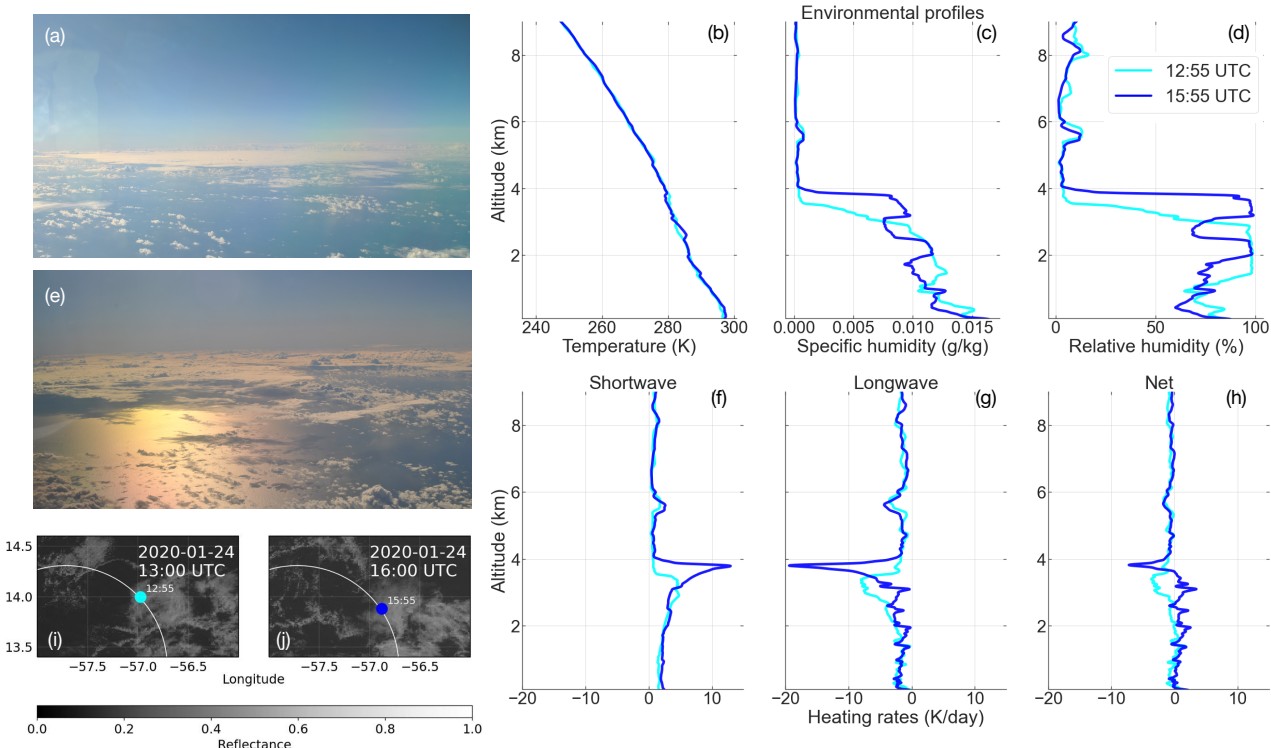

**Figure 6.** Thermodynamic and radiative heating profiles associated with an elevated moisture layer persisting for multiple hours on January 24, 2020 in the HALO flight pattern. Plotted here are the temperature (b), specific humidity (c), relative humidity (d), as well as shortwave, longwave, and net radiative heating rate (f-h) profiles for two soundings sampled three hours apart, at 12:55 and 15:55 UTC. Alongside these profiles are photographs (a,b) taken from the HALO aircraft during the flight and GOES-16 satellite images (i,j), with the dropsonde location and launch time indicated by a circle along the circular flight pattern. Credit for the two flight photographs: J. Vial.

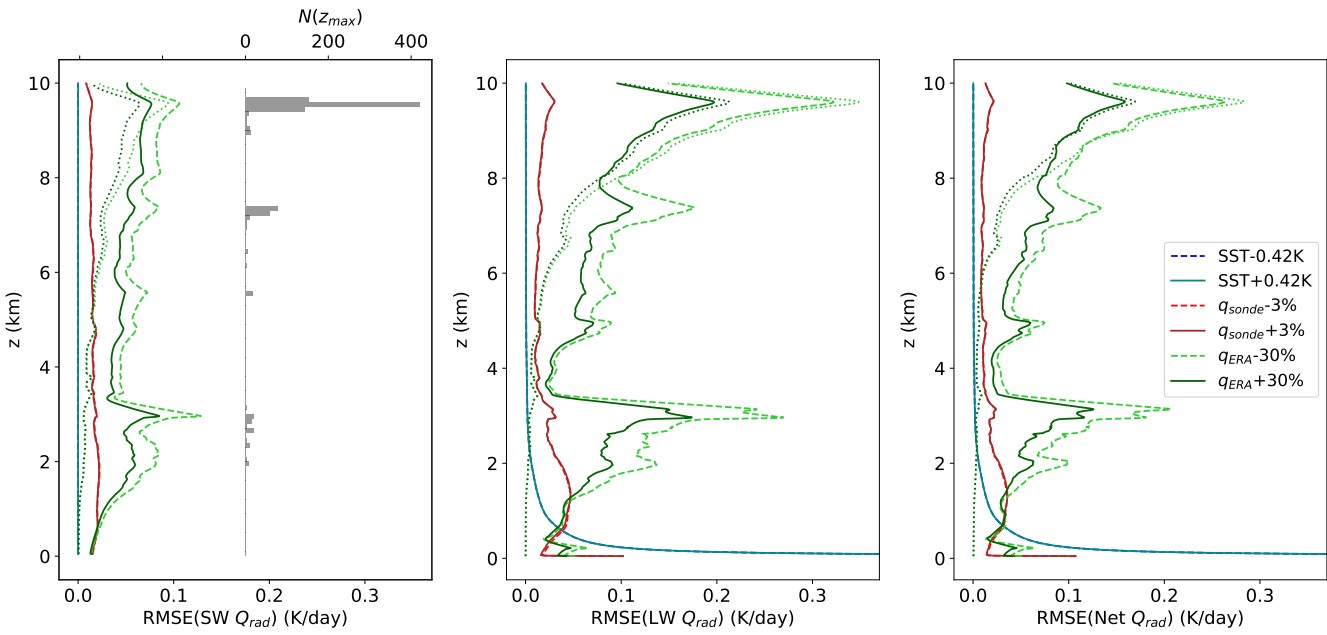

**Figure 7.** Root-mean-square error estimates in shortwave (left), longwave (center) and net heating rates (right) for perturbations in $SST_{skin}$ (blue), ERA5 humidity profiles (green) and sonde humidity measurements (red) for the 1314 daytime profiles that have valid data starting at 40 m. Dashed curves show negative perturbations, solid curves show positive perturbations and dotted green curves show ERA5 humidity perturbations restrained to the 1117 daytime profiles that have valid data at all levels between 40 m and 8 km. The horizontal grey bars on the left panel show the frequency distribution in the maximum level measured ($z_{max}$).