# Peer review of "Atmospheric radiative profiles during EUREC4A"

_Earth System Science Data, 2020_

## Referee Comment (RC1) · Anonymous Referee #1 · 15 Oct 2020

**Title: Atmospheric radiative profiles during EUREC$^4$A**

Authors: Anna Lea Albright Caroline Muller, Benjamin Fildier, Ludovic Touzé-Peiffer, Robert Pincus, Jessica Vial, and Caroline Muller

**General Comments**

This paper describes the creation of an atmospheric radiative dataset based on soundings taken during the EUREC$^4$A field campaign conducted in the trade wind regime near Barbados. The paper is concise, well-written, and does a nice job of conveying how the dataset was created and its importance for understanding the role of radiation in the dynamical and thermodynamical variability of a trade-wind regime. Below are a few comments and questions I have regarding the paper which could use some clarification.

**Specific Comments**

ECMWF analyses have been shown to have biases in different regions of the world (e.g., Nagarajan and Aiyyer 2004). Since you are merging observed sounding profiles with model analyses have the model biases been explored in this region. The paper mentions that the model analyses are merged with the soundings. How is this merging done? Is some type of blending of model and sounding data performed over some vertical layer or do profiles simply switch from observed soundings to model analyses where the sounding data terminate. Such blending may be important especially if significant model biases are present in this region.

In trying to understand how the sounding dataset was constructed and quality controlled, Stephan et al. (2020) briefly mentions the Yoneyama et al. (2002) paper regarding deck heating and cooling effect in sounding data from ships. However, it was unclear if the correction described in Yoneyama's paper was actually applied to the EUREC$^4$A ship sounding data.  Those who work with ship sounding datasets know about these issues and their potential impact on analyses, so please clarify whether or not corrections for deck heating and cooling effects were applied to the sounding datasets. If they were not applied, I would recommend they be done and your radiative computations be revisited.

It would seem that aerosols from the Saharan dust layer could play an important role in radiative computations in this region. However, no mention is made of what, if any, aerosol profiles are used in the radiative computations. Please clarify this information.

Around line 140, it would be helpful if the four different mesoscale organization patterns (Fish, Gravel, Flower, Sugar) could briefly be defined. That is, what distinguishes one pattern from the next.

Even though including clouds in the radiative computations involves ad hoc assumptions, it would be extremely useful to include them in your computations to fully characterize radiative effects in this trade-wind regime. Thus I strongly encourage your team to pursue these computations.

---

## Referee Comment (RC2) · Anonymous Referee #2 · 25 Oct 2020

This paper describes the underlying algorithm and preliminary analysis of a clear-sky radiative heating dataset derived from airborne and ground-based soundings collected during EUREC4A. The paper is well-written and the material is appropriate for ESSD. The dataset introduced here is unique and has clear value for studying the radiative environment around trade cumulus clouds relative to existing satellite products. The methods are robust and applied to a novel regime that is poorly captured by satellites. The dataset is new and not yet described in the literature. The dataset meets the required criteria of being archived and freely available and having a DOI.

The associated analysis is understandably cursory but designed to generate interest in the dataset and demonstrate its utility (see comment above). The authors go as far as to articulate a set of questions for which the dataset may be well-suited. The only

significant weakness is the lack of rigorous uncertainty assessment which is essential for documenting the quality of datasets reported in ESSD. Some justifiable measure of the accuracy of the radiative heating rate profiles is required to allow future users to assess its value for their applications. Currently no quantitative estimates of uncertainty are provided. This shortcoming could likely be resolved through sensitivity studies that simulate the effects of uncertainties in the input datasets (soundings and reanalyses) used to drive them like those employed by others who have produced analogous satellite datasets (e.g. Henderson et al, 2013; Cesana et al, 2019).

Additional Comments:

1. Line 73: A brief discussion of the accuracy of the radiosondes and dropsondes and their implications for errors in computed clear-sky radiative heating should be captured somewhere here. These (in combination with uncertainties in the ERA5 reanalyses) could be used to drive the error required uncertainty analyses. 2. Line 85: It is a little surprising that ERA5 is used to supply SST information. Were no independent SST measurements available during EUREC4A? 3. Line 96: The assumption of clear-skies simplifies the calculations while providing useful information. It would be interesting to generate all-sky profiles in a future study. I do have a minor concern regarding terminology here, though. By convention 'clear-sky' is often adopted to indicate that no clouds were present when the observations were taken. What is actually computed here are 'cloud-free' radiative heating profiles with the understanding that any given profile may or may not have actually contained a cloud. 4. Section 3.2 could be improved to focus on the day-to-day/persistence aspect of the results. As written the primary conclusion that comes across is that the sun comes up and goes down each day. After quickly noting this, it would be better to remove the mean diurnal cycle from Figure 4 and focus on the day-to-day variations in the absence of the DC that's already been covered. The persistence comment on Line 134, for example, is far more interesting than the diurnal cycle. More discussion of this phenomenon as well as the factors that may be responsible for diurnal variations in the LW are warranted. 5. Line 146: Related to (4),

how is the 4-8 hour timescale of trade-wind air masses reconciled with the multi-day persistence noted in Section 3.2?

References:

Henderson, D. S., T. L'Ecuyer, G. Stephens, P. Partain, and M. Sekiguchi, 2013: A multi-sensor perspective on the radiative impacts of clouds and aerosols, J. Appl. Meteor. and Climatol. 52, 853-871.

Cesana, G., D. E. Waliser, T. L'Ecuyer, X. Jiang, and J.-L. Li, 2019: How clouds affect the vertical structure of radiative heating rates: A multi-model evaluation using A-Train satellite observations, J. Climate 32, 1573-1590.

---

## Author Comment (AC1) · 22 Dec 2020

**Atmospheric radiative profiles during EUREC4A**

22 December 2020

We would first like to thank both reviewers for their insightful comments, which we think have substantially improved this manuscript.

We include the reviewer comments and show how we have modified the text in accordance with these suggestions. Changes are also highlighted in color in the revised manuscript, with blue corresponding to Reviewer 1 changes and green for those in response to Reviewer 2. The most substantial addition is an uncertainty assessment section. We quantify the influence of uncertainties in the lower and upper boundary conditions from ERA5 reanalysis, as well as in the humidity profiles from in-situ soundings. Unlike the first submission, we do not filter the soundings anymore based on the levels with valid measurements (so the total size of the dataset is now larger), but we now include two new variables z_min and z_max in the dataset to track the valid range of altitudes measured in the soundings in the final dataset.

Echoing reviewer 2, we do not incorporate the radiative effects of clouds and would leave this to subsequent analysis. Addressing the cloud radiative effect is an important and interesting step, but it would add substantial uncertainty that is beyond the scope of the present manuscript, and we leave it for future work. In this trade-wind cumulus region, the net influence of clouds on the average radiative cooling is modest because of the relatively low observed cloud fraction. Moreover, our original motivation for creating this data set is to investigate how variability in low-level variability in water vapor affects variability in radiative cooling, which, in turn, is thought to drive shallow circulations (e.g. Nigam, 1997, L'Ecuyer et al, 2008, Naumann et al, 2019) and affect the spatial organization of shallow convection (e.g. Muller and Held, 2012). In the abstract and introduction, we have clarified this motivation, as well as underlined that these radiative profiles are cloud-free and aerosol-free profiles.

We hope that the revised manuscript addresses the reviewer comments, but we are happy to consider further revisions. We thank you for your continued interest in this manuscript.

Sincerely,

Anna Lea Albright, Benjamin Fildier, and Ludovic Touzé-Peiffer

**Response to specific comments by Reviewer 1**

1. ECMWF analyses have been shown to have biases in different regions of the world (e.g., Nagarajan and Aiyyer 2004). Since you are merging observed sounding profiles with model analyses have the model biases been explored in this region. The paper mentions that the model analyses are merged with the soundings. How is this merging done? Is some type of blending of model and sounding data performed over some vertical layer or do profiles simply switch from observed soundings to model analyses where the sounding data terminate. Such blending may be important especially if significant model biases are present in this region.

Thank you for requesting this clarification. For upper-level specific humidity and temperature, we switch from in-situ data to ERA5 data at the highest level measured by the sounding. The ERA5 data is interpolated horizontally on the latitude-longitude point of the sounding. The procedure for blending with ERA5 profiles has been clarified in the text around line 85. The changes are highlighted in blue in the manuscript and copied here:

> *The calculation of radiative profiles from radiosonde and dropsonde data then proceeds in the following way:*
>
> 1. *vertical soundings of temperature, pressure, and water vapor specific humidity at 10 meter resolution are interpolated onto a 1 hPa vertical grid and then merged with temperature and specific humidity from ERA5 reanalyses in the following manner. Sonde measurements below 40 m are first truncated for all sondes: radiosondes do not provide data in this surface layer because of deck heating effects on ships (Stephan2020) and we apply the same filter to dropsondes for consistency. The ERA5 profiles at hourly and 0.25 degree resolution are linearly interpolated temporally and spatially to the time, latitude, and longitude of the sounding. ERA5 values are used above the highest level measured by each sonde to extend the observed soundings vertically to 0.1hPa and account for the effect of high-altitude thermodynamic variability on the radiative cooling profiles below. To obtain the lower boundary condition, we linearly interpolate the ERA5 sea surface skin temperature ($SST\textsubscript{skin}$) in the horizontal, also at hourly and 0.25 degree resolution, to the longitude and latitude where the sounding was launched.*

In a new section (4. Uncertainty assessment), we explore sensitivity of the calculated radiative profiles to uncertainties in the sounding and reanalysis data. We describe this section in the response to reviewer 2 below.

As a related test, we sought to assess the ERA5 profiles on the part of the atmosphere where ERA5 and in-situ sounding data overlap. In this vertical region, we find good agreement between the observations and reanalysis, as shown in Fig. 1. Comparing the mean of the soundings and ERA5, the two datasets agree to first-order, with a notable mismatch in the sondes having a sharper decrease in specific humidity between approximately 900 and 750hPa. Over this vertical extent of overlap, the relative consistency between soundings and ERA5, which assimilated the EUREC4A soundings for the EUREC4A period, suggests that biases associated with merging with ERA above the highest level measured by the sonde would be modest.

[Figure]

**Fig 1.** Distribution of soundings and ERA5 data in the HALO circle (approximately 200km-diameter circle centered at -57.7°W and 13.3°N) for the time period of the EUREC4A campaign. This plot serves as a test of whether there is a systematic offset between in-situ observations and reanalysis that would lead to biases when merging these data sets. The means of the soundings (dashed colored lines) and ERA5 (solid black lines) agree well for temperature and moisture. For the same domain and time period, the grey envelope plots all ERA5 profiles, whereas the colored profiles correspond to all in-situ soundings.

2. In trying to understand how the sounding dataset was constructed and quality controlled, Stephan et al. (2020) briefly mentions the Yoneyama et al. (2002) paper regarding deck heating and cooling effect in sounding data from ships. However, it was unclear if the correction described in Yoneyama's paper was actually applied to the EUREC4A ship sounding data. Those who work with ship sounding datasets know about these issues and their potential impact on analyses, so please clarify whether or not corrections for deck heating and cooling effects were applied to the sounding datasets. If they were not applied, I would recommend they be done and your radiative computations be revisited.

Thank you for raising this point. Claudia Stephan told us that these corrections were not applied at level-1. For level-2, the lowest 40 meters of the radiosonde data set were instead deleted to avoid these issues. To standardize the profiles, we have changed the input data set of dropsondes and radiosondes to drop all data below 40m. We perform calculations for all original profiles and include the vertical range of data used for each sonde in new variables z_min and z_max in the final dataset. At the time of first submission, the dropsondes contained data down to 10m, whereas the radiosondes began at 40m above the lower boundary conditions of sea surface temperatures. We do not observe a significant influence of beginning all profiles at 40m above the sea surface temperature.

**We added the following (L87-88) to read:**
*Sonde measurements below 40m are first truncated for all sondes: radiosondes do not provide data because of ``deck heating'' effects (Stephan et al, 2020), and we apply the same filter to dropsondes for consistency.*

3. It would seem that aerosols from the Saharan dust layer could play an important role in radiative computations in this region. However, no mention is made of what, if any, aerosol profiles are used in the radiative computations. Please clarify this information.

Thank you for this important question. We added clarifying text in the paragraph starting around line 105:

> *We compute radiative fluxes and heating rates only for the gaseous component of the atmosphere, without explicitly taking into account cloud or aerosol properties. These radiative profiles are therefore clear-sky and aerosol-free. The soundings do, however, capture the water vapor structure, including regions of high humidity in cloud areas and aerosol layers … Similarly, we do not directly represent the radiative effect of mineral dust aerosols. The dominant aerosol*

*radiative effect in this region has been shown to result from the covariance of aerosols with water vapor, such that aerosols tend to be associated with elevated moisture layers (gutleben2019cloud, gutleben2020radiative). Dust aerosol layers are, moreover, more common in the summer than in winter (lonitz2015signature). We leave open the possibility that direct scattering by dust aerosols has an additional role on radiative heating rates, but do not have the coincident data to appropriately address this question for all soundings.*

4. Around line 140, it would be helpful if the four different mesoscale organization patterns (Fish, Gravel, Flower, Sugar) could briefly be defined. That is, what distinguishes one pattern from the next.

This point is indeed important to clarify, and we are appreciative of this comment. We have added the following text around line 150:

**Before:** We next aggregate radiative heating rates spatially. Fig. 5 illustrates four representative cases of the Fish-Gravel-Flower-Sugar classification established previously for mesoscale organization patterns of clear and cloudy regions (Bony et al, 2020, Stevens et al, 2020).

**Additional text:** *These cloud organization patterns were identified visually from satellite imagery and correspond to differences in large-scale environmental conditions (Bony et al, 2020). They are also observed to have different top-of-the-atmosphere radiative effects (Bony et al, 2020). As outlined in (Stevens et al, 2020), Sugar refers to a `dusting' of small, shallow clouds with low reflectivity and a random spatial distribution. Gravel clouds tend to be deeper than Sugar (up to 3-4km), have little stratiform cloudiness, precipitate, and organize along apparent gust fronts or cold pools at the 20-200km scales. Fish are skeletal networks (often fishbone‑like) of clouds at the 200-2,000km scale with stratiform cloud layers; the Fish pattern is often associated with extratropical intrusions. Flowers are circular features defined by their stratiform cloud elements. Both Fish and Flowers are surrounded by large swaths of clear air.*

5. Even though including clouds in the radiative computations involves ad hoc assumptions, it would be extremely useful to include them in your computations to fully characterize radiative effects in this trade-wind regime. Thus I strongly encourage your team to pursue these computations.

Thank you for this salient comment. We agree that including clouds in the radiative computations is important to fully characterize radiative effects in this trade-wind

regime. We, however, prefer to leave this analysis to a subsequent study for both practical and scientific reasons.

From a practical point of view, we would like to stay as close as possible to the radiosonde and dropsonde data and not add ambiguity with a number of assumptions. We were asked by the second reviewer to better quantify the uncertainty in the underlying data sets. Regarding the in-situ profiles, we simulated the radiative effect of increasing or decreasing the specific humidity profiles by 3%. We believe the present data set would be significantly more difficult to interpret if the effect of clouds was also taken into account, as it would contribute substantially larger uncertainty than result from uncertainties in the underlying sounding, or reanalysis data sets.

From a scientific point of view, one motivation for the present dataset is to investigate how water vapor variability might drive low-level circulations. Prior studies using large-eddy simulations or satellite observations have shown that variability in clear-sky radiative heating drives circulations (e.g. L'Ecuyer et al, 2008, Stephens et al, 2012, Seifert et al, 2015). We seek to provide a new data set based on in-situ observations to further explore these questions.

We nevertheless agree that these motivations should be made more explicit in the manuscript. We therefore chose to change the corresponding paragraph (p. 4) and abstract.

**Changes line around line 100:**

> *We compute radiative fluxes and heating rates only for the gaseous component of the atmosphere, without explicitly taking into account cloud or aerosol properties. These radiative profiles are therefore clear-sky and aerosol-free. The soundings do, however, capture the water vapor structure, including regions of high humidity in cloud areas and aerosol layers. Cloud cover in trade-wind regimes is relatively low, between 10\% (Nuijens2015) and 20\% (Medeiros2016) for active clouds, so cloud-free, or clear-sky, profiles are representative of the thermodynamic environment. Taking into account the influence of cloud liquid water would require a number of ad hoc assumptions about microphysical and optical properties within clouds (see for instance, Guichard2000).*

**Changes to abstract:**
> *We describe the method used to calculate these cloud-free, aerosol-free radiative profiles.*

**Response to general comments by Reviewer 2:**

1. The only significant weakness is the lack of rigorous uncertainty assessment which is essential for documenting the quality of datasets reported in ESSD. Some justifiable measure of the accuracy of the radiative heating rate profiles is required to allow future users to as- sess its value for their applications. Currently no quantitative estimates of uncertainty are provided. This shortcoming could likely be resolved through sensitivity studies that simulate the effects of uncertainties in the input datasets (soundings and reanalyses) used to drive them like those employed by others who have produced analogous satel- lite datasets (e.g. Henderson et al, 2013; Cesana et al, 2019).

Thank you for this insightful comment. We have performed a sensitivity analysis, which we believe is a valuable addition to the present data set. This analysis is described in a new section at the end of the manuscript and illustrated with a new Figure 7.

**See new section "4 - Uncertainty assessment" Line 200**

**Response to specific comments by Reviewer 2**

Additional Comments:

1. Line 73: A brief discussion of the accuracy of the radiosondes and dropsondes and their implications for errors in computed clear-sky radiative heating should be captured somewhere here. These (in combination with uncertainties in the ERA5 reanalyses) could be used to drive the error required uncertainty analyses.

Please see our response to the first general review comment.

2. Line 85: It is a little surprising that ERA5 is used to supply SST information. Were no independent SST measurements available during EUREC4A?

Indeed, as many SST measurements were carried out during EUREC4A (from research vessels, sea gliders, buoys, autonomous platforms, etc.), it may appear surprising that we did not use these SST measurements to supply the lower boundary condition in our calculation of radiative heating rates. There will be a quality-controlled SST dataset that integrates data from all EUREC4A platforms, but it is not yet available, and is not expected to be ready in the coming months. We could update our data set when the EUREC4A SST product becomes available. In the meantime, we employ ERA5 values, interpolated to the latitude and longitude of the sounding, as the lower boundary condition.

This choice is supported by the uncertainty analysis above, as the uncertainty in ERA5-SST does not translate into a large uncertainty in the heating rates profiles except over the first tens of meters of the atmosphere.

Another reason to use ERA5 is that it is the skin temperature, which is relevant for radiative cooling. A number of research vessels also measured sea surface temperature, but a few meters deeper. We see evidence for the offset between the cooler skin temperature and warmer temperature around 2m below in Fig. 2, which compares ERA5 SSTs with those from a research vessel in the same domain, the R/V Meteor. The Pearson correlation between daily mean values (in the same spatial domain and same time period) for ERA5 and R/V Meteor data is r=0.75, demonstrating that these quantities do co-vary, though there is scatter, which could result from the cold skin effect, observational error, that the R/V Meteor data are not final data, or other factors.

[Figure]

Figure 2. Sea surface temperature measurements from ERA5 (red) and R/V Meteor (blue) for the same temporal and spatial domain. The R/V Meteor data are preliminary. ERA5 data correspond to the ocean skin temperature, which is the relevant quantity for radiative heating profiles.

3. Line 96: The assumption of clear-skies simplifies the calculations while providing useful information. It would be interesting to generate all-sky profiles in a future study. I do have a minor concern regarding terminology here, though. By convention 'clear-sky' is often adopted to indicate that no clouds were present when the observations were taken. What is actually computed here are

'cloud-free' radiative heating profiles with the understanding that any given profile may or may not have actually contained a cloud.

Thank you for this clarification request. We have changed the text around line 100 to clarify that these profiles are cloud-free or clear-sky:

*We compute radiative fluxes and heating rates only for the gaseous component of the atmosphere, without explicitly taking into account cloud or aerosol properties. These radiative profiles are therefore clear-sky and aerosol-free. The soundings do, however, capture the water vapor structure, including regions of high humidity in cloud areas and aerosol layers. Cloud cover in trade-wind regimes is relatively low, between 10\% (Nuijens2015) and 20\% (Medeiros2016) for active clouds, so cloud-free, or clear-sky, profiles are representative of the thermodynamic environment.*

4. Section 3.2 could be improved to focus on the day-to-day/persistence aspect of the results. As written the primary conclusion that comes across is that the sun comes up and goes down each day. After quickly noting this, it would be better to remove the mean diurnal cycle from Figure 4 and focus on the day-to-day variations in the absence of the DC that's already been covered. The persistence comment on Line 134, for example, is far more interesting than the diurnal cycle. More discussion of this phenomenon as well as the factors that may be responsible for diurnal variations in the LW are warranted.

We thank the reviewer for this comment about the day-to-day persistence aspect of shortwave, longwave and net heating rates in section 3.2. When conducting the analyses, we had also considered subtracting the mean diurnal cycle in Figure 4 to better show the day-to-day variability. We found the resulting figure, shown below (Figure 3), to be harder to interpret than the original one for two reasons:

1) Since there is some noise in the mean diurnal cycle that we subtract (see for instance the dark blue stripes in the longwave around 5 km at 2 a.m. local time), noisy patterns also appear in the original Figure 4 of our manuscript (see the corresponding red stripes in the figure below around 5 km in the longwave and the net).

2) The colorbar is also slightly difficult to interpret in the new figure below. In this figure below, red regions correspond to more warming relative to the mean diurnal cycle, although it could still correspond to a cooling (e.g. for the longwave/net during the night).

[Figure]

**Figure 3:** Anomalies in heating rates for radiosondes launched BCO over the EUREC4A campaign relative to the diurnal cycle (with the mean diurnal cycle over all soundings subtracted).

Overall, we find the original figure (Figure 4 in the manuscript, also copied below), to be easier to interpret than the modified one. Persistent patterns are more visible and the colorbar is easier to understand. That said, if the reviewers think we should rather include this figure below, with the mean diurnal cycle subtracted, we are of course open to including the altered figure instead.

[Figure]

**Figure 4.** Corresponds to Figure 4 in the first manuscript.

More generally, we agree with the reviewer that the discussion on the day-to-day persistence of patterns observed in Figure 4 warrants improvement. In particular, we did not discuss the link between the radiative heating patterns and variability in temperature and specific humidity, as measured at BCO (shown, for instance, in Stevens et al., 2020, see Figure 5 below). Variations in specific and relative humidity largely explain the day-to-day variability of heating rates, consistent with what we would expect from simple spectral models (e.g. Dopplick et al, 1972, Jeevanjee et al., 2020).

In the text, we propose the following changes in section 3.2:

> *Fig.4 shows the day-to-day evolution of the shortwave (top), longwave (middle) and net (bottom) heating rates derived from radiosondes launched at BCO. In the shortwave and net heating rates, the daily stripes are due to zero shortwave heating during the night. In the longwave component alone, the amplitude of the diurnal cycle is less evident. Regarding the day-to-day variability, both in the shortwave and the longwave components, trends in the height-evolution of the radiative heating maxima appear to persist over several days. \revtwo{These trends are likely due to variations in humidity (e.g. Dopplick1972, Jeevanjee2020) and are consistent with the presence of multi-day trends in moisture observed at BCO during the campaign  (see Figure 13 in Stevens2020). At the end of the campaign, the rise in the peak of longwave cooling appears to correspond to the rising location of the interface between the moist, convecting layer below and dry free troposphere above (not shown). The persistence and evolution of radiative heating patterns could be tied to larger-scale synoptic moisture activity or to the evolution of mesoscale organization patterns.}*

[Figure]

**Figure 5.** Lidar profiling of the lower atmosphere using the CORAL lidar at the BCO. The relative humidity in the lower 5 km is shown over the entirety of the campaign. The

arrow refers to the Lagrangian evolution of humidity, indicative of the magnitude of
vertical velocity variations.  From Stevens et al. (2020).

[Figure]

**Figure 6.** Evolution of relative humidity, specific humidity, and potential temperature for BCO radiosondes from which Figure 4 in our manuscript was calculated.

5. Line 146: Related to (4), how is the 4-8 hour timescale of trade-wind air masses reconciled with the multi-day persistence noted in Section 3.2?

We have removed the phrase about the 4-8 hour timescale from Bony and Stevens, 2019, since it refers to the autocorrelation timescale of large-scale vertical motion, whereas we focus on the radiative effects of thermodynamic conditions. A more complete discussion of variability in radiative heating across scales and the interplay between dynamic and thermodynamic conditions in influencing this radiative variability is beyond the scope of this manuscript, and the phrasing was ambiguous as it was included. Thank you for this comment.

**References:**

Bony, S. and Stevens, B.: Measuring area-averaged vertical motions with dropsondes, Journal of the Atmospheric Sciences, 76, 767–783, https://doi.org/10.1175/JAS-D-18-0141.1, 2019.

Cesana, G., D. E. Waliser, T. L'Ecuyer, X. Jiang, and J.-L. Li, 2019: How clouds affect the vertical structure of radiative heating rates: A multi-model evaluation using A-Train satellite observations, J. Climate 32, 1573-1590.

Dopplick, T. G.: Radiative heating of the global atmosphere, Journal of the Atmospheric Sciences, 29, 278–1294, 240, https://doi.org/10.1175/1520-0469(1972)029<1278:RHOTGA>2.0.CO;2, 1972.

Henderson, D. S., T. L'Ecuyer, G. Stephens, P. Partain, and M. Sekiguchi, 2013: A multi- sensor perspective on the radiative impacts of clouds and aerosols, J. Appl. Meteor. and Climatol. 52, 853-871.

Jeevanjee, N. and Fueglistaler, S.: Simple spectral models for atmospheric radiative cooling, Journal of the Atmospheric Sciences, 77, 479–265497, 2020.

L'Ecuyer, Tristan S., et al. "Impact of clouds on atmospheric heating based on the R04 CloudSat fluxes and heating rates data set." *Journal of Geophysical Research: Atmospheres* 113.D8 (2008).

Muller, Caroline J., and Isaac M. Held. "Detailed investigation of the self-aggregation of convection in cloud-resolving simulations." *Journal of the Atmospheric Sciences* 69.8 (2012): 2551-2565.

Naumann, Ann Kristin, Bjorn Stevens, and Cathy Hohenegger. "A moist conceptual model for the boundary layer structure and radiatively driven shallow circulations in the trades." *Journal of the Atmospheric Sciences* 76.5 (2019): 1289-1306.

Nigam, Sumant. "The annual warm to cold phase transition in the eastern equatorial Pacific: Diagnosis of the role of stratus cloud-top cooling." *Journal of climate* 10.10 (1997): 2447-2467.

Seifert, Axel, et al. "Large-eddy simulation of the transient and near-equilibrium behavior of precipitating shallow convection." *Journal of Advances in Modeling Earth Systems* 7.4 (2015): 1918-1937.

Stephens, Graeme L., et al. "An update on Earth's energy balance in light of the latest global observations." *Nature Geoscience* 5.10 (2012): 691-696.